# R²: A LLM Based Novel-to-Screenplay Generation Framework with Causal Plot Graphs

## Abstract

Automatically adapting novels into screenplays is important for the TV, film, or opera industries to promote products with low costs. The strong performances of large language models (LLMs) in long-text generation call us to propose a LLM based framework Reader-Rewriter (R²) for this task. However, there are two fundamental challenges here. First, the LLM hallucinations may cause inconsistent plot extraction and screenplay generation. Second, the causality-embedded plot lines should be effectively extracted for coherent rewriting. Therefore, two corresponding tactics are proposed: 1) A hallucination-aware refinement method (HAR) to iteratively discover and eliminate the affections of hallucinations; and 2) a causal plot-graph construction method (CPC) based on a greedy cycle-breaking algorithm to efficiently construct plot lines with event causalities. Recruiting those efficient techniques, R² utilizes two modules to mimic the human screenplay rewriting process: The Reader module adopts a sliding window and CPC to build the causal plot graphs, while the Rewriter module generates first the scene outlines based on the graphs and then the screenplays. HAR is integrated into both modules for accurate inferences of LLMs. Experimental results demonstrate the superiority of R², which substantially outperforms three existing approaches (51.3%, 22.6%, and 57.1% absolute increases) in pairwise comparison at the overall win rate for GPT-4o.[1]

## 1 Introduction

Screenplays are the bases of TV, film, or opera-like variants, which are often adapted directly from novels. For example, 52% of the top 20 UK-produced films between 2007-2016 were based on adaptations of novels (Association & Economics, 2018) and the monthly average of TV or movie adaptations in USA for the first nine months of 2024 is more than 10 (Vulture, 2024). Generally, adapting novels into screenplays requires long-term efforts from professional writers. Automatically performing this task could significantly reduce production costs and promote the dissemination of these works (Zhu et al., 2023). However, current work (Zhu et al., 2022; Mirowski et al., 2023; Han et al., 2024; Morris et al., 2023) can only generate screenplays from predefined outlines. Therefore, such an automatic novel-to-screenplay generation (N2SG) is highly expected.

Considering the remarkable performances of large language models (LLMs) in text generation and comprehension tasks (Brown et al., 2020; Ouyang et al., 2022), we are interested in the large language model (LLM) based approach to perform N2SG. However, there are two fundamental challenges ahead before building such a system.

1) How to eliminate the affections of hallucinations in N2SG? Current LLMs like GPT-4 struggle with processing entire novels and often generate various inconsistent contents when processing lengthy input owing to the LLM hallucinations (Liu et al., 2024; Ji et al., 2023; Shi et al., 2023). Existing refinement methods can only roughly reduce such inconsistency and cannot cope with long input data (Madaan et al., 2023; Peng et al., 2023). 2) How to extract effective plot lines capturing the complex causal relationships among events? Eliminating inconsistency alone cannot ensure that generated stories have coherence and accurate plot lines as the original novels. Existing plot graph based methods (Weyhrauch, 1997; Li et al., 2013) depict plot lines in the linearly ordered events.

---

[1]Code and data will be released later.

Figure 1: Typical rewriting process of a human screenwriter. A screenwriter needs *read* and *rewrite* multiple times with iterative refinements when adapting the novel to a screenplay.

However, events may be intricate and intertwined, and those methods cannot model the complex causalities.

For the first challenge, we could just part-by-part refine the context associated with the inconsistency caused by hallucinations. Therefore, a hallucination-aware refinement method (HAR) is proposed in this work to iteratively eliminate the affections of LLM hallucinations for better information extraction and generation from long-form texts.

For the second one, the plot lines including causalities should be extracted for coherent rewriting. Plot graphs are convenient to represent sequential events and can be extended as causal plot graphsto embed the causalities. Therefore, a causal plot-graph construction (CPC) is proposed in this article to robustly extract the causal relationships of events with the causal plot graphs.

Now the question is how to build an N2SG system with HAR and CPC. Looking at human screenwriters (Figure 1), we see that they can successfully do it by a reading and rewriting composed procedure (McKee, 1999): First, they *read* the novels to extract the key plot events and character profiles (*i.e.*, character biographies and their relationships) for constructing plot lines of the novels; Then, they *rewrite* the novels into screenplays according to those plot lines which are adapted into the story lines and scene goals as outline guiding the script writing. Both reading and rewriting steps may apply multiple times with multiple refinements until satisfaction is achieved.

Inspired by the iterative-refinement based human rewriting process, we propose the Reader-Rewriter ($R^2$) framework (Figure 2). The Reader adopts a sliding window based strategy to scan the whole novel by crossing the chapter bounds, so that events and character profiles can be effectively captured for the following CPC process to build the causal plot graph, in which HAR is deployed to extract accurate events and character profiles. The Rewriter adopts a two-step strategy to first obtain the storylines and goals of all scenes as global guidance and then generate the screenplay scene by scene under the precise refinement from HAR, ensuring coherence and consistency across scenes.

Experiments on $R^2$ are conducted on a test dataset consisting of several novel-screenplay pairs and the evaluation is based on the proposed seven aspects. The GPT-4o-based evaluation shows that $R^2$ significantly outperforms the existing approaches in all aspects and gains overall absolute improvements of 51.3%, 22.6%, and 57.1% over three compared approaches. Human evaluators similarly confirm the strong performances of $R^2$, demonstrating its superiority in N2SG tasks.

In summary, the main contributions of this work are as follows:

1) Hallucination-aware refinement method (HAR) for refining the LLM outputs, which can eliminate the inconsistencies caused by the LLM hallucinations and improve the applicability of LLMs.

2) A causal plot-graph construction method (CPC), which takes a greedy cycle-breaking algorithm to extract the causality embedded plot graphs without cycles and low-strength relations of events.

3) A LLM based framework $R^2$ for N2SG, which adopts HAR and CPC, and mimics the human screenplay rewriting process with the Reader and Rewriter modules for the automatically causal-plot-graph based screenplay generation.

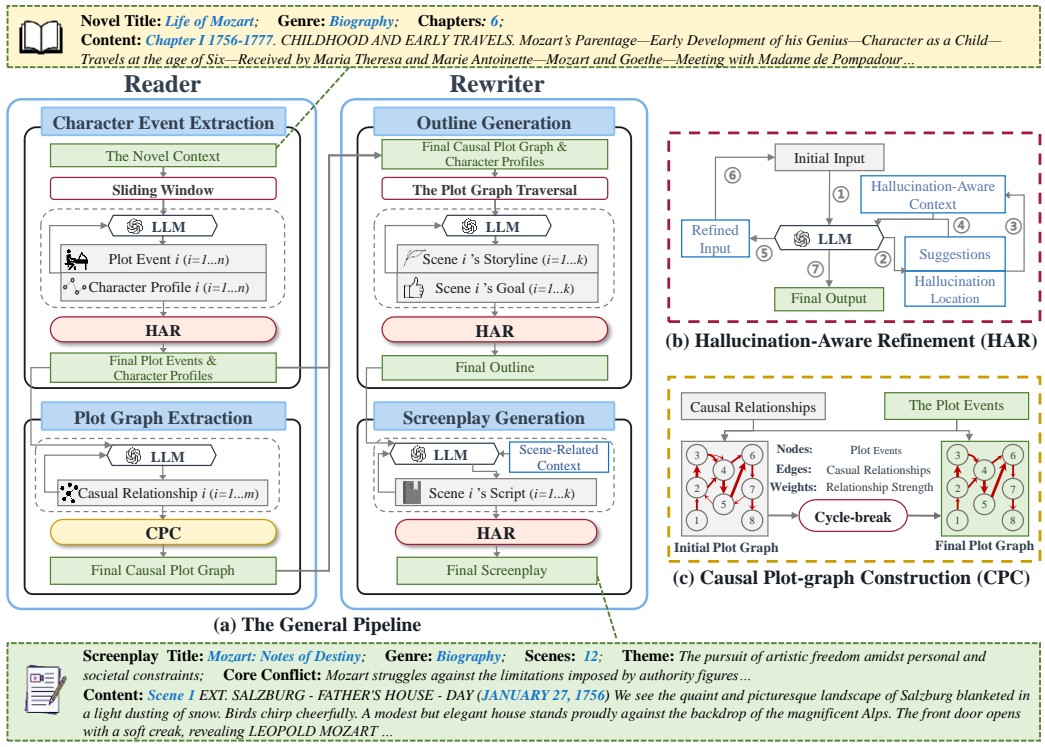

Figure 2: Structure of the Reader-Rewriter ($R^2$). The general pipeline (a) of $R^2$ consists of two modules, the Reader and the Rewriter, where two strategies, the Hallucination-Aware Refinement (HAR) (b) and the Causal Plot-graph Construction (CPC) (c) are integrated to efficiently utilize LLMs and understand the plot lines. The arrows indicate data flow between the different modules. The examples in the figure are for better illustration.

## 2 FOUNDATIONS FOR LLM BASED NOVEL-TO-SCREENPLAY GENERATION

There are two challenges for the LLM based N2SG. First, the LLM outputs can be quite different from the expected ones owing to the hallucinations. Consequently, LLMs may extract and generate non-existent events and screenplays. Second, understanding the plot lines of novels is very important to generate coherent and consistent screenplays. Plot graphs are often used to describe the plot lines, which should capture the complex causalities among events. For the first challenge, the hallucination-aware refinement meth (HAR) is introduced, so that the affections of LLM hallucinations can be significantly mitigated (Sec. 2.1). For the second challenge, a causal plot-graph construction method is proposed to efficiently build the causalities embedded plot graphs (Sec. 2.2).

### 2.1 HALLUCINATION-AWARE REFINEMENT

HAR prompts the LLM to identify the intrinsic inconsistencies caused by the hallucinations, locate where the hallucinations occur in the LLM outputs, and provide suggestions for refinement.

Denote the LLM as $\mathcal{M}$. In the round $t$, HAR (Figure 2 (b)) first identifies the hallucination locations $loc_t$ where the intrinsic inconsistencies occur in the input $x_t$ and generates suggestions $sug_t$ describing how $\mathcal{M}$ refines them. Then the hallucination-aware context $c_t$ is extracted from the input and corresponding support texts based on the hallucination locations, and input to $\mathcal{M}$ to refine the hallucination part in $x_t$ as $r_t$. Next, $r_t$ is merged into $x_t$ as $x_{t+1}$ for the $t+1$-th round of self-refinement. This self-refinement process continues until the initial input data is fully processed and consistent, culminating in the refined output. Algorithm 1 presents the full process of HAR.

---

**Algorithm 1** Hallucination-aware refinement

---

**Require:** Initial input $x_0$, support text $s$, LLM $\mathcal{M}$, prompts $\{p_{\text{fb}}, p_{\text{refine}}\}$, stop condition $\text{stop}(\cdot)$, context retrieve function $\text{retrieve}(\cdot)$, task-specific prompt $p_{\text{fb}}$, input-output-feedback refined quadruple examples $p_{\text{refine}}$.
1: **for** iteration $t \in \{0, 1, \dots\}$ **do**
2:     $loc_t, sug_t \leftarrow \mathcal{M}(p_{\text{fb}} \parallel x_t)$            ▷ Locate the hallucinations and provide suggestions
3:     $c_t \leftarrow \text{retrieve}(loc_t, s)$                  ▷ Retrieve the hallucination-aware context
4:     **if** $\text{stop}(loc_t, sug_t, t)$ **then**
5:        **break**                                 ▷ Check the stop condition
6:     **else**
7:        $r_t \leftarrow \mathcal{M}(p_{\text{refine}} \parallel c_t \parallel sug_t)$        ▷ Get the refined part of input
8:        $x_{t+1} \leftarrow \text{Merge } r_t \text{ into } x_t$        ▷ Update the input with the refined part
9:     **end if**
10: **end for**
11: **return** refined and consistent output $x_t$

---

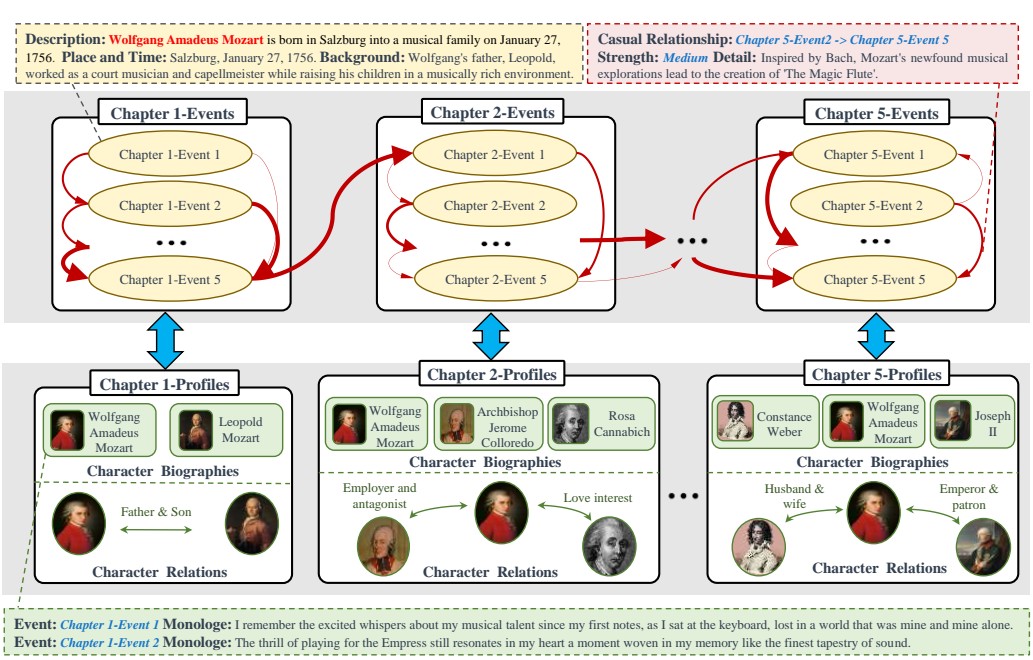

Figure 3: Demonstration of the causal plot graphs and character profiles. The plot lines are represented as directed acyclic graphs (DAGs) composed of events and their causal relationships. Thicker arrows represent stronger relations.

## 2.2 CAUSAL PLOT-GRAPH CONSTRUCTION

**The Causal Plot Graph** The causal plot graph (Figure 3) embeds causalities of events by graphs. Here *causal* means the graph is built according to the critical connections between key events, not just their sequences in novels. Especially, this type of plot graph is designed as a directed acyclic graph (DAG) for the causal relationships (edges) between the plot events (nodes).

Formally, the causal plot graph is a tuple $G = \langle E, D, W \rangle$, where $E$ denotes the events composed of place and time, background, and description; $D$ describes the causal relationships of events; and $W$ indicates the strengths of causal relationships, classified into three levels: *High* for direct and significant influences; *medium* for partial or indirect influences; and *low* for minimal or weak influences.

**The Greedy Cycle-breaking Algorithm** The initial extracted plot graph by LLMs often contains cycles and low-strength relationships owing to the LLM hallucinations. Therefore, a variant of Prim's algorithm (Prim, 1957) is proposed to remove these cycles and unimportant relationships. Called as the greedy cycle-breaking algorithm, it breaks cycles based on the relationship strength and the degree of event node.

Specifically, the causal relations are first ordered by their weights $W$ as $D$ from high level to low one. Two relations of the same level will be set to a lower sum of the degrees of its connected event endpoints, so that more important edges between more important nodes are prioritized. Denote each directional relation $d(a, b)$ with $a$ and $b$ being the start and end points of $d$ respectively. The forward reachable set of endpoint $x$ is $S_x$. If the end $b$ of $d \in D$ is already reachable to its start $a$ via previously selected causal relation edge, $d$ is skipped to avoid forming a cycle. Otherwise, it is added to the edge set $F$, and $S_x$ of all endpoints of the edges in $F$ is updated to reflect the new connections, ensuring the set $F$ remains acyclic. As a result, the set $F$ forms the edge set of the directed acyclic graph (DAG) that preserves the most significant causal relationships while preventing cycles. Algorithm 2 gives the details of this algorithm.

---

**Algorithm 2** Greedy cycle-breaking algorithm for causal plot-graph construction

---

**Require:** $E$: set of plot events, $D$: set of causal relations, $W$: set of relation strengths.
1: Sort $D$ by $W$ (from high to low) and the sum of endpoints degrees (from less to more)
2: **for** each edge $d \in D$ and its endpoints $a, b$, where $a, b \in E$ **do**
3:    **if** $a \in S_b$ **then**
4:       **continue**                                 ▷ Skip if $a$ is reachable from $b$
5:    **end if**
6:    Add $d$ to $F$                                 ▷ Add to acyclic edge set
7:    Update $S_x$ of each endpoint $x$ for all edges in $F$    ▷ Update the forward reachable sets
8: **end for**
9: **return** $F$ as the causal relation edge set of the DAG

---

Now we discuss R$^2$ based on the proposed two fundamental techniques.

## 3 THE PROPOSED READER-REWRITER FRAMEWORK

R$^2$ (Figure 2 (a)) consists of two main components according to the human rewriting process, the Reader and the Rewriter. The former extracts the plot events and character profiles, and constructs the causal plot graphs, while the latter adapts the novels into screenplays with the graphs and profiles.

### 3.1 LLM-BASED READER

The LLM-based Reader takes two sub-modules: The character event extraction and the plot graph extraction.

**Character Event Extraction** The Reader first identifies the plot events from the novel and extracts them in a chapter-by-chapter way because of the limited input context window of LLMs. Here the LLMs extract event elements such as description, place, and time (Figure 3). This is implemented by prompting LLMs to generate structured outputs (Bi et al., 2024).

To better cope with long texts of novels, a sliding window based technique is first introduced during event extraction. Sliding through the full novel with a chapter-sized window, this strategy ensures the extracted events consistent across chapters. It is also applied to extract character profiles in each chapter (Figure 2). Then HAR (Sec. 2.1) is taken to reduce the inconsistencies in plot events and character profiles caused by LLM hallucinations. Here, the LLM is recursively prompted to identify the inconsistencies and refine them according to the relevant chapter context, so that the inconsistencies between the events and profiles are significantly reduced.

**Plot Graph Extraction** The extracted events are utilized to further construct the causal plot graphs by the proposed CPC method. Specifically, firstly the LLM is recursively prompted to identify the new casual relationship according to the relevant chapter context. After the graph is connected and

no new relationships are added to it, CPC is performed to eliminate the cycles and low-weight edges in the graph, so that the obtained causal graphs can more effectively and accurately reflect the plot lines in the novels.

## 3.2 LLM-BASED REWRITER

The Rewriter is organized into two subsequent steps: The first step is to create the screenplay outlines of all scenes with the second for iteratively generating the screenplay of each scene. Those two steps are packed as two corresponding sub-modules: The outline generation and the screenplay generation. The final screenplay is iteratively refined by HAR.

**Outline Generation**  A screenplay adaptation outline can be constructed with the plot graph and character profiles (Figure 2), which consists of the story core elements, the screenplay structure, and a writing plan including the storyline and goal for each scene. Three different methods are used to traverse the plot graphs, depth-first traversal (DFT), breadth-first traversal (BFT), and the original chapter order (Chapter), corresponding to three different screenplay adaptation modes, *i.e.*, adapting the screenplay based on the main storyline (depth-first), the chronological sequence of events (breadth-first), or the original narrative order of the novel.

The misalignment of events and characters often happens during the outline generation, especially when generating the scene writing plans. Therefore, $R^2$ performs HAR (Sec. 2.1) to get the initial screenplay adaptation outlines. This process focuses on the alignment of key events and major characters and returns the final adaptation outlines.

**Screenplay Generation**  Now each scene can be written based on its writing plan (Figure 2) which includes the storyline, goal, place and time, and character experiences. The LLM is prompted to generate each scene with the scene-related context which consists of the relevant chapter and the previously generated scene. Then HAR verifies whether the generated scene meets the storyline goals outlined in the writing plan. This approach ensures the consistency between the generated screenplay scenes and maintains alignment with the related plot lines of the novels.

## 4 EXPERIMENTS

### 4.1 EXPERIMENTAL SETTING

**Dataset and Evaluation**  A novel-to-screenplay dataset was created by manually cleaning pairs of novels and screenplays collected from public sources to evaluate the performance of N2SG. The novels are categorized into popular and unpopular groups based on their ratings and number of reviews. To ensure fairness, both types are included in the testing sets. Such dataset will be open for future research of both trainable and train-free applications.

To ensure a balanced assessment, the proposed $R^2$ method adopts five novels—two from the popular category and three from the unpopular category as testing set, and no training samples are involved. To further minimize subjective bias caused by reading long texts at one time, we select a total of 15 excerpts in novels for every human evaluator, with each excerpt limited to around 1000 tokens.

In the evaluation, 15 human evaluators are employed to focus on seven aspects, including *Interesting*, *Coherent*, *Human-like*, *Diction and Grammar*, *Transition*, *Script Format Compliance*, and *Consistency*. The pairwise comparisons through questionnaires are designed. Their responses were then aggregated to compute the win rate (Equation 1) for each aspect. However, since human evaluators often exhibit large variances in their judgments, GPT-4o[2] is also utilized as the main evaluator to give the judgment according to the same questionnaires. This can enhance objectivity and reduce potential bias in the results. Appendix A presents further details of the dataset and evaluation.

**Task Setup**  The $R^2$ framework uses the optimal parameters obtained from the analysis experiments (Sec. 4.4) with the refinement round set to 4 and the plot graph traversal method set to BFT for comparing with the competitors. It employs GPT-4o-mini[3] with low-cost and fast inference as

---

[2]https://platform.openai.com/docs/models/gpt-4o
[3]https://platform.openai.com/docs/models/gpt-4o-mini

the backbone model, since our target is to build an effective and practical N2SG system. During inference, the generation temperature is set to 0 for reproducible and stable generations.

**Compared Approaches** $R^2$ is compared against ROLLING, Dramatron (Mirowski et al., 2023), and Wawa Writer[4]. ROLLING is a vanilla SG method that generates 4,096 tokens at a time via GPT-4o-mini using the $R^2$-extracted plot events and all previously generated screenplay text as prompt. Once the generation arrives at 4,096 tokens, it will be added to the prompt for iteratively generating the screenplay. Dramatron is an approach that generates screenplays from loglines. Here we input the $R^2$-extracted plot events to it for comparison. Wawa Writer is a commercially available AI writing tool, whose novel-to-screenplay features are adopted for performance comparison.

Table 1: Comparison of $R^2$ in the win rate against three approaches evaluated by GPT-4o (%).

| Approach | Interesting | Coherent | Human-like | Dict & Gram | Transition | Format | Consistency | Overall |
|---|---|---|---|---|---|---|---|---|
| ROLLING | 19.2 | 34.6 | 26.9 | 15.4 | 30.8 | 15.4 | 23.1 | 24.4 |
| $R^2$ | **80.8** (↑61.6) | **65.4** (↑30.8) | **73.1** (↑46.2) | **84.6** (↑69.2) | **69.2** (↑38.4) | **84.6** (↑69.2) | **76.9** (↑53.8) | **75.6** (↑51.3) |
| Dramatron | 39.3 | 46.4 | 35.7 | 42.9 | 28.6 | 35.7 | 50.0 | 39.3 |
| $R^2$ | **60.7** (↑21.4) | **57.1** (↑10.7) | **64.3** (↑28.6) | **57.1** (↑14.2) | **71.4** (↑42.8) | **64.3** (↑28.6) | **57.1** (↑7.1) | **61.9** (↑22.6) |
| Wawa Writer | 10.7 | 32.1 | 25.0 | 10.7 | 25.0 | 35.7 | 21.4 | 22.0 |
| $R^2$ | **89.3** (↑78.6) | **75.0** (↑42.9) | **75.0** (↑50.0) | **89.3** (↑78.6) | **75.0** (↑50.0) | **64.3** (↑28.6) | **78.6** (↑57.1) | **79.2** (↑57.1) |

Table 2: Comparison of $R^2$ in the win rate against three approaches evaluated by human (%).

| Approach | Interesting | Coherent | Human-like | Dict & Gram | Transition | Format | Consistency | Overall |
|---|---|---|---|---|---|---|---|---|
| ROLLING | 35.9 | 40.1 | 36.6 | 19.0 | 35.2 | 35.2 | 45.1 | 34.5 |
| $R^2$ | **71.8** (↑35.9) | **66.9** (↑26.8) | **73.9** (↑37.3) | **83.1** (↑64.1) | **70.4** (↑35.2) | **88.7** (↑53.5) | **77.5** (↑32.4) | **74.9** (↑40.4) |
| Dramatron | 40.0 | 47.8 | 48.9 | **61.1** | 47.8 | 48.9 | **66.7** | 50.6 |
| $R^2$ | **74.4** (↑34.4) | **52.2** (↑4.4) | **54.4** (↑5.5) | 40.0 (↓21.1) | **56.7** (↑8.9) | **77.8** (↑28.9) | 55.6 (↓11.1) | **57.4** (↑6.9) |
| Wawa Writer | 43.8 | 40.0 | 47.5 | 45.0 | 43.8 | 47.5 | 45.0 | 44.4 |
| $R^2$ | **62.5** (↑18.7) | **67.5** (↑27.5) | **62.5** (↑15.0) | **62.5** (↑17.5) | **62.5** (↑18.7) | **60.0** (↑12.5) | **50.0** (↑5.0) | **62.1** (↑17.7) |

## 4.2 MAIN RESULTS

The quantitative comparison in Table 1 shows that $R^2$ consistently outperforms the competitors (overall, 51.3% gain for Rolling, 22.6% gain for Dramatron, and 57.1% gain for Wawa Writer). In particular, $R^2$ demonstrates clear superiority in *Dict & Gram* (69.2% gain for Rolling) and *Interesting* (78.6% gain for Wawa Writer). These results demonstrate that $R^2$ can generate linguistically accurate and fantastic screenplays with smooth transitions. Moreover, human evaluation results in Table 2 demonstrate $R^2$ overall outperforms its counterparts across most aspects, especially in *Interesting* and *Transition*, indicating its ability to generate fantastic and fluent screenplays. Only compared to Dramatron, $R^2$ has a slightly poor performance in *Dict & Gram* and *Consistency*. A possible reason is the human preference for the long-form narrative generated by Dramatron.

The qualitative analysis for the generated screenplays indicates the following disadvantages of the compared approaches: Owing to the lack of iterative refinement and limited understanding of the plots of novels, the screenplays generated by the ROLLING often perform poorly compared to $R^2$ in the *Interesting*, *Transition*, and *Consistency* aspects. Dramatron tends to generate screenplays similar to drama, frequently generating lengthy dialogues, which leads to poor performance in the *Interesting*, *Format*, and *Transition* aspects. As for Wawa Writer, the screenplays it generates frequently demonstrate plot inconsistencies between scenes and *Diction and Grammar* issues, indicating its backbone model may lack of deep understanding of the novel.

## 4.3 ABLATION STUDY

This study assesses the effectiveness of relevant techniques by GPT-4o (Table 3). First, removing the HAR led to a significant drop in *Dict & Gram* (38.4% lose) and *Consistency* (46.1% lose), showing that HAR is critical to enhance language quality and consistency. Second, removing the

---

[4]https://wawawriter.com

CPC causes a significant drop in *Interesting* (64.2% lose) and *Consistency* (71.4% lose), indicating that CPC is essential in generating fantastic and consistent screenplay. Finally, excluding all context supports results in a sharp decrease in *Transition* (66.6% lose), *Consistency* 77.8% lose), indicating its importance in improving plot transitions and consistency.

Table 3: Ablation results of $R^2$ in win rate evaluated by GPT-4o.

| Approach | Interesting | Coherent | Human-like | Dict & Gram | Transition | Format | Consistency | Overall |
|---|---|---|---|---|---|---|---|---|
| $R^2$ | **61.5** | **61.5** | **65.4** | **69.2** | **61.5** | **61.5** | **76.9** | **77.7** |
| w/o HAR | 38.5($\downarrow$23.0) | 38.5 ($\downarrow$23.0) | 38.5($\downarrow$26.9) | 30.8($\downarrow$38.4) | 38.5($\downarrow$23.0) | 46.2($\downarrow$15.3) | 30.8 ($\downarrow$46.1) | 44.7 ($\downarrow$33.0) |
| $R^2$ | **82.1** | **64.3** | **78.6** | **71.4** | **82.1** | **78.6** | **85.7** | **92.1** |
| w/o CPC | 17.9 ($\downarrow$64.2) | 85.7 ($\uparrow$21.4) | 21.4($\downarrow$57.2) | 28.6 ($\downarrow$42.8) | 28.6 ($\downarrow$53.5) | 64.3 ($\downarrow$14.3) | 14.3($\downarrow$71.4) | 44.3($\downarrow$47.8) |
| $R^2$ | **66.7** | **77.8** | **77.8** | **66.7** | **83.3** | **88.9** | **88.9** | **92.2** |
| w/o Context | 33.3 ($\downarrow$33.4) | 50.0 ($\downarrow$27.8) | 22.2 ($\downarrow$55.6) | 33.3($\downarrow$33.4) | 16.7 ($\downarrow$66.6) | 11.1 ($\downarrow$77.8) | 11.1($\downarrow$77.8) | 33.3 ($\downarrow$58.9) |

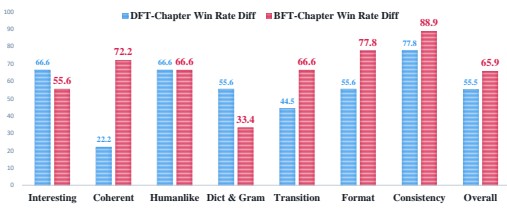

(a) Analyse of different traversal methods

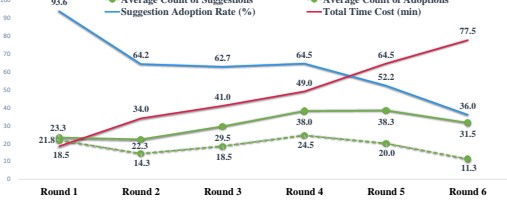

(b) Analyse of different refinements rounds

Figure 4: The effect of traversal method and refinement rounds.

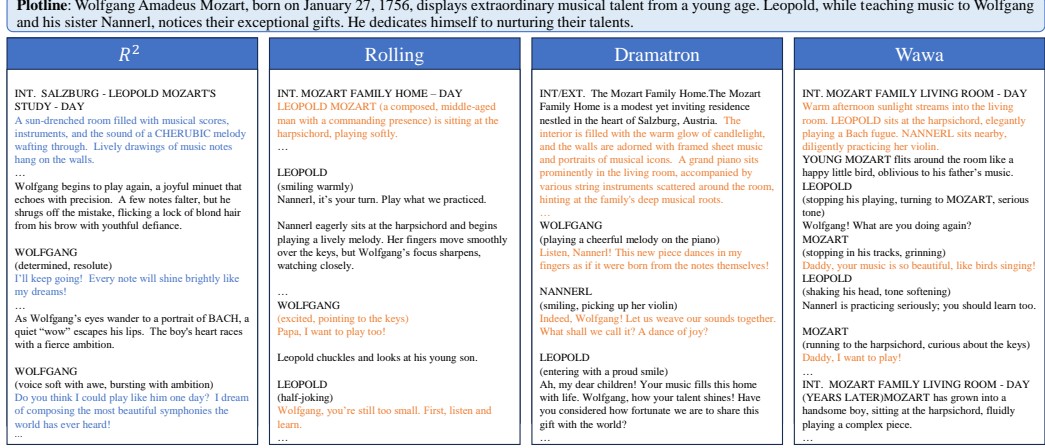

Figure 5: The case study on the different approaches.

## 4.4 EFFECT OF DIFFERENT FACTORS

**The Plot Graph Traversal Methods** The effect of different plot graph traversal methods on screenplay adaptation is explored (Figure 4 (a)). Here the win rate difference compared to Chapter is directly exhibited since Chapter's performance is behind the other methods. Overall, BFT outperforms DFT and demonstrates significant advantages in *Coherent*, *Transition*, *Format*, and *Consistency*. This illustrates BFT's effectiveness for telling complex stories with intertwined plots, while DFT maintains strong performance in creating fantastic stories. These results confirm that BFT offers the best balance for plot coherence and overall quality in screenplay adaptation.

**The Rounds of Refinements** Figure 4 (b) demonstrates when the number of refinement rounds increases, the number of suggestions rises in the first four rounds and then begins to decline, indicating that there is less room for improvement. The suggestion adoption rate shows a downward trend,

stabilizing around 60% during rounds 2 to 4, with a noticeable drop in round 5. Moreover, the time cost is progressively higher as the refinement rounds increase. Therefore, four refinement rounds achieve the best balance between refinement quality and efficiency.

## 4.5 CASE STUDY

A case study is also undertaken to demonstrate the effectiveness of $R^2$, where the screenplay segment generated by $R^2$ and three approaches are presented in Figure 5. $R^2$ outperforms other approaches in creating vivid settings, expressive dialogue, and integrating music with character development. For instance, $R^2$ effectively enhances the mood through the elegant scene setting and emphasizes Wolfgang's passion and ambition through the emotional dialogue. These elements make the screenplay more immersive and emotionally driven compared to simpler treatments in other scripts.

## 5 RELATED WORK

**Long-form Generation** Recently, many studies (Yang & Klein, 2021; Yang et al., 2022; Lei et al., 2024) have emerged on long-form text generation with LLM, which aim at solving challenges include long-range dependency issues, content coherence, premise relevance, and factual consistency in long-form text generation, *etc.* $Re^3$ (Yang et al., 2022) introduces a four-stage process (plan, draft, rewrite, and edit) for long story generation, using recursive reprompting and revision; DOC (Yang et al., 2023) focuses on generating stories with a detailed outline linked to characters and uses a controller to ensure coherence and control. Compared to those multi-stage generation frameworks driven by the story outline, our approach uniquely leverages a condensed causal plot graph and character profiles for automatic and consistent screenplay generation from novels.

Other work focuses on constructing human-AI collaboration frameworks for screenplay generation (Zhu et al., 2022; Mirowski et al., 2023; Han et al., 2024; Zhu et al., 2023). Dramatron (Mirowski et al., 2023) presents a hierarchical story generation framework that uses prompt chaining to guide LLMs for key screenplay elements, building a human collaboration system for long-form screenplay generation. IBSEN (Han et al., 2024) allows users to interact with the directors and character agents to control the screenplay generation process. These studies emphasize collaborative, multi-agent approaches with human-LLM interactions. In contrast, our approach solves N2SG by automatically generating long-form screenplays from novels, minimizing the user involvement.

**LLM-Based Self-Refine Approach** Iterative self-refinement is a fundamental feature of human problem-solving (Simon, 1962). LLMs can also improve the quality of their generation through self-refinement (Madaan et al., 2023; Saunders et al., 2022; Scheurer et al., 2024; Shinn et al., 2023; Peng et al., 2023; Madaan et al., 2023). LLM-Augmenter (Peng et al., 2023) uses a plug-and-play module to enhance LLM outputs by incorporating external knowledge and automatic feedbacks. Self-Refine (Madaan et al., 2023) demonstrates that LLMs can improve their outputs across various generation tasks by multi-turn prompting. In this paper, $R^2$ utilizes the LLMs-based self-refinement approach to tackle challenges in causal plot graph extraction and long-form text generation.

## 6 CONCLUSION

This paper introduces a LLM based framework $R^2$ for the novel-to-screenplay generation task (N2SG). Two techniques are first proposed, a hallucination-aware refinement (HAR) for better exploring LLMs by eliminating the affections of hallucinations and a causal plot-graph construction (CPC) for better capturing the causal event relationships. Adopting those techniques while mimicking the human rewriting process leads to the Reader and Rewriter composed system for plot graph extraction and scene-by-scene screenplay generation. Extensive experiments demonstrate that $R^2$ significantly outperforms the competitors. The success of $R^2$ establishes a benchmark for N2SG tasks and demonstrates the potential of LLMs in adapting long-form novels into coherent screenplays. Future work could explore integrating control modules or multi-agent frameworks into $R^2$ to impose more stringent constraints and expand it to broader long-form story generation tasks to further develop the capability of our framework.

ETHICS STATEMENT

Our study uses publicly available data and does not involve human subjects or sensitive data. We ensure that no ethical concerns, such as bias, unfairness, or privacy issues, arise from our work. There are no conflicts of interest or legal issues related to this research, and all procedures comply with ethical standards.

REPRODUCIBILITY STATEMENT

We ensure the reproducibility of our results by providing comprehensive descriptions of our models, datasets, and experiments. The source code and data processing steps are made available as supplementary material.

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

## A  DATASET AND EVALUATION DETAILS

The details of evaluation dataset are shown in Table 4.

Table 4: Details of the experimental dataset. The column headers represent the following: **Size** refers to the number of the novel-script pairs; **Avg.Novels** represents the average words of novels; **Avg. Screens** indicates the average words of screenplays; **Reviews** denotes the average adapted movie reviews; **Rating** refers to the average adapted movie rating; **Genres** indicates the categories of adapted movie genres.

| Dataset Type | Size | Avg.Novels | Avg.Screens | Reviews | Rating | Genres |
|---|---|---|---|---|---|---|
| Test | 5 | 86,482 | 32,358 | 269,770 | 7.6 | Action / Suspense / Crime / Biography / Sci-Fi |
| Unpopular | 10 | 159,487 | 28,435 | 62,527 | 6.5 | Suspense / Crime / Comedy / Love / Romance / Sci-Fi / Adventure / Thriller / Biography / History / Drama |
| Popular | 10 | 133,996 | 29,737 | 757,127 | 8.98 | Sci-Fi / Thriller / Drama / Suspense / Action / Crime / War / Biography / History |

**Evaluation Methods**   Similar to the prior work such as Re[3] (Yang et al., 2022) and DOC (Yang et al., 2023), pairwise experiments are conducted by designing questionnaires and presenting them to human raters. Each questionnaire consists of an original novel excerpt or a logline (depending on the competitors), two screenplay excerpts (denoted as $A$ and $B$, with random ordering), and a set of questions evaluating seven aspects (Table 5). Each aspect includes one to two questions, with control questions taken to ensure accuracy in the responses. Each survey question has only four options: $A$, $B$, or both are good, or neither good.

Table 5: Evaluation criteria for screenplay

| Criterion | Description |
|---|---|
| *Interesting* | Degree of capturing the interest of readers. |
| *Coherent* | Degree of the smooth development of plots and scene transitions. |
| *Human-like* | Language quality resembling human writing. |
| *Diction and Grammar* | Accuracy of word choice and grammar. |
| *Transition* | Degree of the natural flow of the story and emotional shifts between scenes. |
| *Script Format Compliance* | Adherence to the screenplay formatting rules. |
| *Consistency* | Degree of the consistency with the original novel plot. |

Evaluators are recruited and training is provided before completing the questionnaires. Each evaluator must read the original novel, compare the screenplay excerpts $A$ and $B$, and answer the survey based on the comparison. The evaluators are not informed of the sources of the screenplays and are instructed to select the option that best aligns with their judgment. Finally, the questionnaire results are aggregated and the win rate (WR) of a screenplay $X \in \{A, B\}$ for each aspect $i$ is computed by the formula:

$$\text{WR}_{X,i} = \frac{N_{X,i} + N_{AB,i}}{N_T \times Q_i} \tag{1}$$

where: $N_{X,i}$ is the number of evaluators who prefer to screenplay $X$ in aspect $i$; $N_{AB,i}$ is the number of evaluators who found both screenplay $A$ and $B$ suitable in aspect $i$; $N_T$ is the total number of evaluators; $Q_i$ is the number of questions in aspect $i$.

**The Consistency of Evaluators**   Cohen's kappa coefficient is used to measure the consistency of opinions between two evaluators when filling out the questionnaire. The value of Cohen's kappa ranges from -1 to 1, with 1 indicating complete agreement, 0 for the same consistency as a random selection, and a negative value for a lower consistency than a random selection. There are three different evaluators by random selection in each of the three groups of experiments. The average value of Cohen's kappa for every two evaluators on all questionnaires are calculated, and the Cohen's

kappa heat maps in the comparative experiments with rolling (Figure 6a), Dramatron (Figure 6b), and Wawa (Figure 6c) are obtained. Among these three figures, the highest Cohen's kappa is only 0.44, and there are even negative Cohen's kappa values between three pairs of evaluators, which clearly shows that the consistency of opinions between evaluators is quite low.

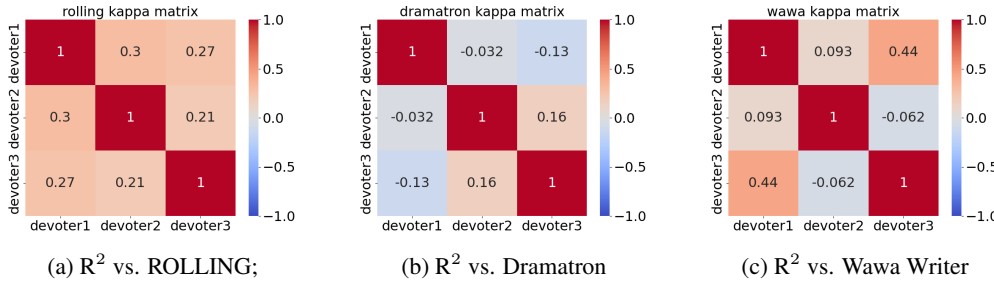

(a) $R^2$ vs. ROLLING;                (b) $R^2$ vs. Dramatron                (c) $R^2$ vs. Wawa Writer

Figure 6: Cohen's kappa heat map between three evaluators in each comparison questionnaire.

