# OpenReview forum: "R$^2$: A LLM Based Novel-to-Screenplay Generation Framework with Causal Plot Graphs"
_ICLR.cc/2025/Conference — ICLR 2025 Conference Withdrawn Submission_

### Official Review · Reviewer_K1zp · 2024-10-27

**Soundness:** 2
**Presentation:** 3
**Contribution:** 1
**Rating:** 3
**Confidence:** 4

**Summary:**

This paper introduces a novel framework for generating screenplays from novels using causal plot graphs, aiming to enhance the coherence and reduce hallucinations in the generated screenplays. The authors present a hallucination-aware refinement (HAR) method to minimize inconsistencies in large language model (LLM) outputs, resulting in more reliable screenplay quality. Additionally, causal plot graphs are employed to maintain logical and causal coherence throughout the screenplay. The paper is well-structured, with comprehensive experiments including both AI (GPT-4o) and human evaluations, as well as case studies to demonstrate the effectiveness of the proposed approach.

**Strengths:**

**Originality & Significance**

1.	This paper proposes a new novel-to-screenplay generation framework with causal plot graphs to achieve automatic and high-quality generated screenplay with reduced hallucination.

2.	This work introduces a hallucination-aware refinement method (HAR) to eliminate the hallucination in the LLM output for a more consistent screenplay, which significantly enhances the quality of the generated screenplay.

3.	This work utilizes causal plot graphs to generate screenplay that is more coherent and consistent.

**Clarity & Quality**

1.	The paper is well-structured and easy to follow. The task is well-formulated.
2.	Substantial experiments are conducted to show the effectiveness of the proposed method. Both AI evaluator (GPT-4o) and human evaluators are involved to evaluate the proposed method, which is comprehensive and convincing.
3.	Moreover, the authors conduct case study to analyze how the proposed method gives the high-quality screenplay.

**Weaknesses:**

W1. Though this work aims to address the current two issues in LLM-based novel-to-screenplay generation method, the significance of developing novel-to-screenplay applications might be a bit questionable. With the advent of generative AI, it is more promising for generative AI to directly generate screenplay instead of transforming from novels.

W2. The paper lacks novelty. The two key technical contributions of this work are HAR and CPC. However, HAR simply adopts the self-refinement idea which has been widely studied. Moreover, CPC includes causal graph identification and cycle removal, where the cycle removal simply use the existing algorithms and the more crucial identification is not sufficiently discussed and introduced.

W3. The method lacks necessary illustration of how LLMs are prompted, especially for the causal plot graph, which is a key concept throughout the whole paper.

**Questions:**

1.	Regarding W1, the supportive figures in the paper are from 2007 to 2016. The need for novel-to-screenplay seems less convincing.

2.	For W2, what is the difference between the proposed HAR and the existing self-refinement approaches in other areas (e.g., NLP) to reduce hallucination?

3.	For the CPC, how to make sure the identified causal relationships between events are reliable? Is HAR also applied to the causal relationship generation process?

4.	Why are the results of the proposed methods different across comparisons between different methods (e.g., results of R^2 are all different across Table 1-3). Did I miss anything important regarding experimental settings?

5.	What are the prompts for LLMs used in this paper? These are important to achieve reproducibility of this work.

---

### Official Review · Reviewer_KmoQ · 2024-11-02

**Soundness:** 2
**Presentation:** 3
**Contribution:** 2
**Rating:** 3
**Confidence:** 4

**Summary:**

The paper proposes R2 (Reader-Rewriter), a framework that uses LLMs to automatically convert novels into screenplays. It addresses two main challenges: LLM hallucinations and complex plot causality relationships through two key innovations - HAR (Hallucination-aware refinement) and CPC (Causal plot-graph construction). The framework consists of a Reader module that extracts plot events and a Rewriter module that generates screenplays, showing significant performance improvements over existing approaches in experimental evaluations.

**Strengths:**

1. Takes a systematic approach to handling LLM hallucinations, a common problem in long-text generation.
2. Mimics human screenwriting processes, making the approach more interpretable and potentially more reliable
3. Well written and structured

**Weaknesses:**

1. Lack of experimental results for open source models.
2. Lack of related work. [1]
3. Lack of discussion about potential copyright or ethical implications of automated screenplay adaptation.
4. The paper is essentially a prompt engineering work focusing on NOVEL-TO-SCREENPLAY conversion. While this application has practical significance, it may not align well with ICLR's scope. The work would be more suitable for CL or HCI venues, as it lacks substantial theoretical machine learning innovations.
5. As an HCI-oriented screenplay generation application, the paper lacks comprehensive human evaluation protocols and user studies. The paper doesn't provide sufficient details about how human evaluators managed to assess such lengthy content effectively and consistently. The methodology for user testing and practical usability assessment is notably absent


[1] HoLLMwood: Unleashing the Creativity of Large Language Models in Screenwriting via Role Playing

**Questions:**

Refer to Weaknesses.

---

### Official Review · Reviewer_Zc36 · 2024-11-02

**Soundness:** 2
**Presentation:** 2
**Contribution:** 2
**Rating:** 3
**Confidence:** 4

**Summary:**

This paper introduces $R^2$, an LLM-based framework for converting novels into screenplays. It addresses two main challenges: managing LLM hallucinations and capturing causal plot structures.  Experiments show that  $R^2$ outperforms other methods in generating coherent and high-quality screenplays.

**Strengths:**

1. $R^2$ features two challenges and employs Hallucination-Aware Refinement (HAR) for consistency and Causal Plot-graph Construction (CPC) to model event relationships;
2. The framework is tested against multiple baselines, highlighting its superiority in linguistic accuracy and plot coherence.

**Weaknesses:**

1. The paper’s writing is difficult to follow, especially regarding the relationship between key concepts like “Reader” and “Rewriter” modules versus “HAR” and “CPC.” The connections between these two groups could be clarified for better understanding.
2. Experiments are only conducted on specific LLMs (i.e., GPT family), which may limit $R^2$ applicability to other models or require adaptation for other LLMs.

**Questions:**

1. How does this paper specifically handle cross-chapter events?
2. Besides the final generated screenplay results, what are the individual effects of HAR and CPC?

---

### Note · Authors · 2024-11-15

I have read and agree with the venue's withdrawal policy on behalf of myself and my co-authors.